# Bilateral Facial Weakness with Distal Paresthesia Following COVID-19 Vaccination: A Scoping Review for an Atypical Variant of Guillain–Barré Syndrome

**DOI:** 10.3390/brainsci12081046

**Published:** 2022-08-07

**Authors:** Yoo-Hwan Kim, Jee-Eun Kim, Byeol-A Yoon, Jong-Kuk Kim, Jong-Seok Bae

**Affiliations:** 1Department of Neurology, Hallym University Sacred Heart Hospital, Anyang 14068, Korea; 2Department of Neurology, Ewha Womans University College of Medicine, Seoul 03760, Korea; 3Department of Neurology, Dong-A University College of Medicine, Busan 49201, Korea; 4Peripheral Neuropathy Research Center, Dong-A University College of Medicine, Busan 49201, Korea; 5Department of Neurology, Kangdong Sacred Heart Hospital, Seoul 05355, Korea

**Keywords:** Guillain–Barré syndrome, COVID-19 vaccine, adverse effects, facial paralysis, systematic review

## Abstract

**Background and Purpose:** Recent population-based studies from the US and UK have identified an increase in the occurrence of Guillain–Barré syndrome (GBS) following coronavirus disease 2019 (COVID-19) vaccination. However, the localized variant of GBS might be underestimated due to its rarity and atypical features. We aimed to identify and characterize bilateral facial weakness with distal paresthesia (BFWdp) as a GBS variant following COVID-19 vaccination. **Materials and Methods:** Relevant studies published during the COVID-19 pandemic were searched and identified in the MEDLINE, Embase, and other databases. **Results:** This review found that 18 BFWdp cases presented characteristics similar to previous BFWdp cases as defined in the literature: male dominance, frequent albuminocytological dissociation, and acute inflammatory demyelinating neuropathy pattern. In contrast, facial nerve enhancement on brain MRI and antiganglioside antibody positivity were often observed in BFWdp following COVID-19 vaccination. **Conclusions:** The mechanism of BFWdp following COVID-19 vaccination appears to be somewhat different from that of sporadic BFWdp. Neurological syndromes with rare incidence and difficulty in diagnosis should be considered adverse events of COVID-19 vaccination.

## 1. Introduction

The coronavirus disease 2019 (COVID-19) pandemic induced the rapid development and administration of vaccines. This is a historically unprecedented event, and vaccine-related adverse effects (AEs) have been the main concerns of both the public and clinicians. Guillain–Barré syndrome (GBS) is one of the most frequently considered AEs in this context, and its occurrence after COVID-19 vaccination has been considered nationwide [1,2,3]. Despite differences in registration methods, country, and case definition, these nationwide studies found that viral vector vaccine administration resulted in a significant increase in GBS incidence [1,2,3].

The incidence of localized GBS variants as an AE of COVID-19 vaccination can be underestimated due to its atypical or relatively mild symptoms following vaccination. Recent population-based studies from the US and UK could only analyze GBS codes that were confined to the classic GBS outcomes of admission or death [1,2,3]. It can be assumed that those studies excluded relatively mild GBS or localized GBS variants from consideration as vaccine AEs.

Bilateral facial weakness with distal paresthesia (BFWdp) is a rare GBS variant, has been nosologically defined by consensus, and occurs in less than 1% of GBS cases [4]. Recent studies (mostly case reports or case series) have consistently identified patients with subsequent BFWdp or similar events after COVID-19 vaccination [5,6,7]. Therefore, the incidence of localized GBS could be underestimated, and it would be a distinct finding if BFWdp cases are frequently observed after COVID-19 vaccination.

This scoping review aimed to identify and characterize BFWdp following COVID-19 vaccination and thereby provide pathological insight into the occurrence of this rare syndrome following COVID-19 vaccination.

## 2. Materials and Methods

### 2.1. Data Sources and Study Selection

On 30 May 2022, we identified relevant studies through electronic searches of medical subject headings and keyword searches of MEDLINE (PubMed) and Embase using the following terms: ‘Guillain-Barré Syndrome’, ‘variant’, ‘cranial nerve diseases’, ‘facial palsy’, ‘bilateral facial palsies’, ‘facial diplegia’, ‘distal paresthesia’, ‘bifacial weakness’, ‘COVID-19’, ‘SARS-CoV-2’, ‘COVID-19 vaccines’, ‘ChAdOx1 nCoV-19’, ‘AstraZeneca vaccine’, ‘2019-nCoV vaccine mRNA-1273’, ‘Moderna vaccine’, ‘BNT162 vaccine’, ‘Pfizer and BioNTech vaccine’, ‘Baiya SARS-CoV-2 VAX COVID-19 vaccine’, ‘Sinovac COVID-19 vaccine’, ‘Ad26COVS1’, and ‘Janssen vaccine’. The reference lists of the selected articles were systematically reviewed for other potentially relevant citations.

### 2.2. Final Enrollment of Studies and Data Extraction

Two researchers (Y.H.K. and J.-E.K.) independently curated titles and abstracts. In case of disagreement, a consensus on the articles of which the full texts should be screened was reached through discussion. The same two researchers then independently screened full-text articles for inclusion or exclusion, following the same procedure. We designed a data extraction form to collect age (in years), sex, comorbidities, type of vaccine administrated (AstraZeneca, Moderna, Pfizer, or other type), vaccine dose (first or second), preceding infection (upper respiratory infection, diarrhea, or others), interval between vaccination and initial symptoms (in days), first subjective symptoms, presence of typical and atypical BFWdp features (bilateral facial palsies and distal paresthesia), cerebrospinal fluid (albuminocytological (A/C) dissociation), findings of nerve conduction studies (NCS), brain MRI findings, and antiganglioside antibody assay results.

Three authors (Y.H.K., J.-E.K., and J.S.B.) used the extracted data from the eligible studies, with discrepancies resolved through discussion. The following exclusion criteria were applied: (1) unclear definition of BFWdp due to a lack of clinical or laboratory information; (2) insufficient differential diagnosis from BFWdp-mimicking diseases, such as idiopathic cranial neuropathies of other causes (i.e., sarcoidosis or Lyme disease); or (3) GBS with definite weakness or ataxia of the limbs, including classic GBS with a sole initial manifestation of bilateral facial weakness. The final inclusion of studies was based on the agreement of all the authors.

### 2.3. Standard Protocol Approval, Registration, and Patient Consent

This systematic review was based on bibliometric data without animal or human data, and so it was not necessary to obtain ethical approval.

## 3. Results

The search strategy identified 48 potential articles. We also searched articles that cited or were referenced in any of the initially identified articles. After removing duplicates and reviewing titles/abstracts, the full texts of ten articles were read. Of these, three articles were excluded: two because classic GBS presented with bilateral facial weakness initially or during the disease course, and one because bilateral facial weakness was studied, which is not compatible with a GBS variant diagnosis. Seven articles were finally included in this systematic review; however, some cases reported in three articles were also excluded. Despite the authors diagnosing BFWdp, we only included the cases among those articles that met our enrollment criteria [5,8]. For example, one article reported five BFWdp cases, ref [7], but we excluded three of them because of symptoms in the limbs: right hip flexion weakness, sensory ataxia, and bilateral hip flexion weakness. Finally, 18 cases from seven articles were analyzed in this review [5,6,7,8,9,10,11]. A flow diagram illustrating the full enrollment process is shown in Figure 1.

Table 1 lists a summary of the finally enrolled studies. Four studies were case series, and three were case reports. Two were from the UK, two were from the US, two were from Italy, and one was from Argentina. The study from Argentina analyzed the largest number of cases (nine BFWdp cases); however, it provided limited information on the clinical course, such as initial symptoms and their subsequent progression [6].

Table 2 lists the demographic, clinical, and laboratory characteristics of 18 BFWdp cases following COVID-19 vaccination. The cases were aged 53.2 ± 14.2 years (mean ± SD). There were more males among the finally enrolled cases (14/18, 78%). The vaccine types were as follows: AstraZeneca (11/18, 61%), Sputnik V (4/18, 22%), and Janssen (2/18, 11%). No case received the Pfizer or Moderna vaccines. All but one of the cases presented A/C dissociation (17/18, 94%). Available information from NCS of limbs indicated either acute inflammatory demyelinating polyneuropathy (AIDP) pattern (7/9) or normal findings (2/7). Brain MRI performed in four cases revealed facial nerve enhancement. In the study from Argentina, antiganglioside antibody tests identified antiganglioside antibodies in three cases (two with IgG anti-GM1 and one with anti-GD1a) and antisulfatide antibodies (one case). One of those cases died from sudden arrhythmia onset.

## 4. Discussion

Simultaneous bilateral facial nerve palsy is rare and can have various causes. In view of GBS variants, Susuki et al. [12] reported on 22 patients with clinical signs such as acute progressive bilateral facial weakness, paresthesia in the distal limbs, and hypo- or areflexia. However, cranial dominant and classic GBS often initially manifest with bilateral facial weakness and subsequent sensory and motor symptoms in the limbs. Some suspected BFWdp cases following COVID-19 vaccination have been this type of GBS, [13,14,15,16,17,18,19] and so unequivocal motor weakness or ataxia of the limbs during the disease course negated BFWdp diagnoses. Therefore, we defined BFWdp in this analysis as a bilateral facial weakness with concomitant distal paresthesia without major dysfunction of limbs (i.e., unequivocal weakness or prominent ataxia).

Based on a careful definition of BFWdp, we identified BFWdp variants following COVID-19 vaccination. The 18 enrolled cases were nosologically compatible with the definition from the previous consensus [4]. The BFWdp variant is known to be very rare, with one prospective study finding that this variant only accounted for about 1% of all GBS cases [4]. Despite its rarity, cases in the present review had characteristics compatible with the previous definition of BFWdp variants [4,12,20]: male dominance, frequent A/C dissociation, and AIDP pattern. In contrast, the cases in the present study often presented with facial nerve enhancement on brain MRI and antiganglioside antibody positivity.

GBS after COVID-19 vaccination could be related to the generation of host antibodies that cross-react with proteins in peripheral myelin or axons. These antibodies might be generated in direct response to the SARS-CoV-2 spike protein. In addition, a secondary immune response involving components of the adenovirus vector was also proposed. However, the development of BFWdp during COVID-19 infection suggests a direct immunological response to the spike protein [5]. This provides evidence of the SARS-CoV-2 spike protein binding to sialic-acid-containing glycoproteins and gangliosides on cell surfaces, which increases viral transmissibility [21]. Antibody cross-reactivity between the SARS-CoV-2 spike protein and peripheral nerve glycolipids or gangliosides may be involved in the pathogenesis of GBS associated with SARS-CoV-2 infection or vaccination. In another aspect, the specific genetic background and leucocyte antigen haplotype profile of the host may also play a role, as it does in GBS associated with SARS-CoV-2 and other autoimmune neurological disorders [22].

Most cases of GBS following vaccination have been related to viral vector vaccines such as those from AstraZeneca or Janssen [1,2,3]. GBS has historically been considered in influenza vaccination programs. A swine influenza vaccine program resulted in an increase of about sevenfold in GBS cases (A/New Jersey/1976/H1N1) in the US [23]. Thereafter, in the early 1990s, the GBS incidence after influenza vaccination programs was associated with a 1.7-fold increase in the risk of GBS relative to the unvaccinated population [24]. Despite the relatively clear distinction between classic GBS and GBS following vaccination, we could not conclude whether variants of BFWdp are common or related to any special immunological processes after COVID-19 vaccination. However, the frequent reporting of this rare variant is a distinct feature of the COVID-19 era.

It is presumed that 70–80% of the population has active COVID-19 immunity thanks to infection or vaccination, which breaks the disease chain [25]. Additionally, the duration of the effective period of the COVID-19 vaccination mandates the administration of regular doses or booster shots of the vaccine. This suggests that AEs following COVID-19 vaccination may continue, and so clinicians should pay more attention to any neglected AEs after COVID-19 vaccination.

We should address several limitations of this scoping review: first, because of the different medical environments in the centers reporting each case, there might exist a concern of diagnostic certainty–availability of neurology specialists, electrophysiological studies, or special testing such as antiganglioside antibody assays. In addition, some of the enrolled cases could be suspected as other atypical variants or overlapped types of GBS. Finally, a causal link between vaccination and the development of BFWdp variants could not be confirmed by the observational studies included in this analysis.

This scoping review has highlighted that a neurological disease with a low incidence and difficulty in diagnosis should be considered an AE of COVID-19 vaccination. Studies targeting the epidemiological and scientific features of the disease across the COVID-19 pandemic and endemic era can provide useful clues for the understanding of neuroimmunological diseases such as GBS or other rare variants. In addition, better characterization of any severe adverse effects associated with COVID-19 vaccines allows the general population to get a better idea of the risks associated with the COVID-19 vaccine and hopefully reduce vaccine hesitancy [26].

## Figures and Tables

**Figure 1 brainsci-12-01046-f001:**
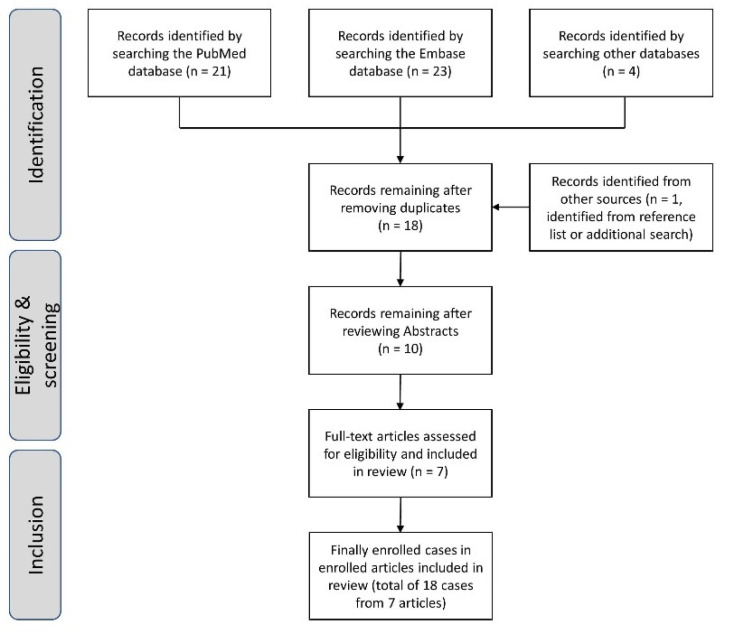
Flow diagram demonstrating the inclusion/exclusion process for studies included in the final analyses.

**Table 1 brainsci-12-01046-t001:** Summary of enrolled studies.

First Author. Reference	Country	Journal	Year	Study Type	No. of Patients	Case No.
Nasuelli [9]	Italy	Neurological Sciences	2021	Case report	1	[1]
Jain [10]	US	Cureus	2021	Case report	1	[2]
Allen [5]	UK	Annals of Neurology	2021	Case series	3/4 (cases 1, 2, and 4)	[3,4,5]
Rossetti [11]	US	Journal of Emergency Medicine	2021	Case report	1	[6]
Andreozzi [8]	Italy	Neurological Sciences	2022	Case series	1/2 (case 1)	[7]
Bonifacio [7]	UK	Journal of Neurology, Neurosurgery, and Psychiatry	2022	Case series	2/5 (cases 1 and 5)	[8,9]
Castiglione [6]	Argentina	Neuromuscular Disorders	2022	Case series	9/9	[10,11,12,13,14,15,16,17,18]

**Table 2 brainsci-12-01046-t002:** Demographic, clinical, and laboratory characteristics of patients with bilateral facial weakness with distal paresthesia following COVID-19 vaccination.

Case No.	Age (Years)	Sex	Vaccine Type	Dose	Comorbidities	Interval between Vaccination and Initial Symptoms (Days)	Initial Subjective Symptoms	BFW	Distal Paresthesia	Other Findings	A/C Dissociation	NCS Findings (Limbs)	Antiganglioside Antibodies	Microbiology Test for Lyme Disease
[1]	59	M	AZ	First	HT, hyperuricemia	10	Distal paresthesia	+	+	Postural instability	+	AIDP	-	NA
[2]	65	F	Janssen	First	HT, DM, DL, drug allergy	15	Pain, ageusia, hyposalivation	+	-	Dysarthria, dysphagia, dysphasia	+	NA	NA	NA
[3]	54	M	AZ	First	-	12	Distal paresthesia	+	+	-	+	Normal	NA	NA
[4]	20	M	AZ	First	UC	20–21	Headache, distal paresthesia	+	+	-	+	Normal	NA	NA
[5]	55	M	AZ	First	HT	22–23	Thigh paresthesia, lumbosacral numbness	+	?	Facial nerve enhancement on MRI	+	NA	NA	NA
[6]	38	M	Janssen	First	Anxiety, depression, marijuana user	12–14	Distal (limbs, tongue, lip) paresthesia, numbness, facial palsy	+	+	Facial nerve enhancement on MRI, hyponatremia	+	NA	NA	NA
[7]	59	F	AZ	First	Hashimoto thyroiditis	15	Lower limbs and back paresthesia, pain, BFP	+	+	-	+	AIDP	NA	NA
[8]	66	M	AZ	First	-	7	Back/leg pain, distal paresthesia	+	+	Facial nerve enhancement on MRI	+	AIDP	-	-
[9]	53	M	AZ	First	-	8	Lower back discomfort and radiating pain	+	+	-	+	NA	NA	-
[10]	56	F	Sputnik V	First	NA	19	NA	+	+	-	-	Normal	-	NA
[11]	55	M	Sputnik V	First	NA	28	NA	+	+	-	+	Normal	-	NA
[12]	87	M	Sputnik V	First	NA	17	NA	+	+	Sudden arrhythmia onset, death	+	Normal	GD1a (+)	NA
[13]	50	M	AZ	First	NA	20	NA	+	+	-	+	Normal	-	NA
[14]	39	M	Sputnik V	First	NA	10	NA	+	+	-	+	NA	Sulfatide (+)	NA
[15]	42	F	AZ	First	NA	28	NA	+	+	-	+	Normal	NA	NA
[16]	52	M	AZ	First	NA	13	NA	+	+	-	+	AIDP	GM1 (+)	NA
[17]	43	M	Sputnik V	Second	NA	13	NA	+	+	-	+	AIDP	-	NA
[18]	65	M	AZ	Second	NA	13	NA	+	+	Facial nerve enhancement on MRI	+	Normal	GM1 (+)	NA

A/C, albuminocytological; AIDP, acute inflammatory demyelinating polyneuropathy; AZ, AstraZeneca; BFW, bilateral facial weakness; DL, dyslipidemia; DM, diabetes mellitus; F, female; HT, hypertension; M, male; NA, not available; NCS, nerve conduction study; UC, ulcerative colitis.

## Data Availability

The datasets generated or analyzed during the study are available from the corresponding author on reasonable request.

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
