# Peer review of "Bilateral Facial Weakness with Distal Paresthesia Following COVID-19 Vaccination: A Scoping Review for an Atypical Variant of Guillain–Barré Syndrome"

_brainsci, 2022, doi:10.3390/brainsci12081046_

Round 1

Reviewer 1 Report

The present paper tackles a very important topic on vaccine AEs. We all know that vaccines are among the most important discoveries of modern medicine, however in potential adverse reactions and their impact on public remain underestimated precisely due to the crucial advantages of public vaccinations. 

The introduction of the paper provides sufficient amount of background data, however the purpose of the study is not clearly described and can be improved. It took me 5-6 times reading to understand exactly what the authors hypothesized. 

Authors hypothesize that that mild cases of a rare GBS form are often masked and remain underreported and overlooked when investigating COVID-19 vaccine. They investigate their hypothesis through reviewing case reports of neurological conditions that could potentially match their hypothesis. This is an example of confirmation bias in my opinion. While such review could be a first step into looking for underreported cases of rare GBS form it certainly cannot draw the conclusions that the authors have drawn as those are speculative. 

From technical point of view - English language level is good, extra intervals are present here and there and need to be removed. 

"Johnson and Johnson's" and "Janssen" are used interchangeably in the text whereas consistency is necessary. "Johnson's and Johnson's Janssen COVID-19 vaccine" is the full name of the product which acknowledges that this is a product of Janssen, a Johnson and Johnson's company. 

Reviewer 2 Report

The authors have done an interesting work and presented it in well defined way. Overall manuscript is good and have a significant importance. However, I have several concerns which need to be addressed before the publication.  

1. Line no. 40: What is the meaning of GBS variant?

Please explain the relevance of the variant 

I wonder, what this GBS variant actually means, To increase the readability this needs top be explained in the introduction

In one of the article, it is written like

reported case of the facial diplegia variant of GBS which makes more sense. I recommend to change it to exclude the confusions. 

Introduction must be improved while providing recent references of the COVID-19 articles like

http://dx.doi.org/10.1080/21645515.2022.2045853

Burckhardt RM, Dennehy JJ, Poon LLM, Saif LJ, Enquist LW. Are COVID-19 Vaccine Boosters Needed? The Science behind Boosters. J Virol. 2022 Feb 9;96(3):e0197321. doi: 10.1128/JVI.01973-21. Epub 2021 Nov 24. PMID: 34817198; PMCID: PMC8827020.

The need of vaccines (http://dx.doi.org/10.1080/21645515.2022.2045853) and the booster doses can be explained (https://doi.org/10.1128/jvi.01973-21) and then can be associated with GBS and can be explained further.

 2. Line no. 67: Abstracts to abstracts. 

3. Somewhere after the Discussion , there must be conclusion section which gives the highlights or major findings of the study. I will increase the reach of the manuscript.,

4. Additionally, future directions can be provided.

5. In the discussion, the tone must be neutral which will not strictly say that the vaccines will have adverse effects as the probability is significantly lower.

Additionally, there is no any statistical model so it is better to stay neutral. 

6. Furthermore I suggest to incorporate more recent relevant data in the discussion to justify the results. This will increase the credibility of the study.

Reviewer 3 Report

Dear editor,

Thank you for the kind invitation to review this manuscript.

The authors described an interesting scoping review with regards to atypical variant of Guillain-Barre syndrome after COVID-19 vaccination. The study is generally well written and easy to follow. 

Attached below are my comments for the author's consideration

Methodology

- The methods should be written as per Prisma scoping review checklist

Results

- Are there any possibility of obtaining the Naranjo scoring for likelihood of adverse effects being attributed to COVID-19 vaccine. 

Discussion

- What are the main implications of the study results?

- What are the limitations of this study?

Minor comments

- To substantiate the need for this study, it is probably important to correlate it with high vaccine hesitancy noted in studies

-> Better characterisation of any severe adverse effects associated with COVID-19 vaccine allows the general population to get a better idea of the risk of COVID-19 vaccine and hopefully reduce vaccine hesitancy

-> Relevant citations: https://pubmed.ncbi.nlm.nih.gov/34452026/; https://www.frontiersin.org/articles/10.3389/fpubh.2021.770985/full

Reviewer 4 Report

Thank  you for permitting me to review this manuscrript 

In this review the authors have assessed the possible effect of covid 19 vaccination on  incidence BFW dp  a minor  localied  entity  part  of GBS 

The authors provided the incidence of bfwdp in general population before Covid 19 pandemic which was about 1% of all GBS, they need also to provide that estimated  incidence in the vacinated population 

The authors should provide the final defintion which was used to determine BFWDP in this review  line 134

Line 143 does it mean that the other case had not facial nerve enhancement in MRI 

Please elaborate  speculations about antibodies and cross reactions of proteins line 146

Line 148 PPR  (please provide reference )

Line 159 PPR

I think it would be better remove the GBS from since after  the authors do not attribute this AE to GBS

Round 2

Reviewer 1 Report

I am satisfied with the response and the addition of study limitations is satisfying. I would recommend this article for publication with the editions.